# Persistence of Long COVID Symptoms Two Years After SARS-CoV-2 Infection: A Prospective Longitudinal Cohort Study

**DOI:** 10.3390/v16121955

**Published:** 2024-12-20

**Authors:** Gili Joseph, Ili Margalit, Yael Weiss-Ottolenghi, Carmit Rubin, Havi Murad, Raquel C. Gardner, Noam Barda, Elena Ben-Shachar, Victoria Indenbaum, Mayan Gilboa, Sharon Alroy-Preis, Yitshak Kreiss, Yaniv Lustig, Gili Regev-Yochay

**Affiliations:** 1The Sheba Pandemic Preparedness Research Institute (SPRI), Sheba Medical Center, Tel Hashomer, Ramat Gan 52621, Israel; gili.joseph@sheba.health.gov.il (G.J.); ili.margalit@sheba.health.gov.il (I.M.); yael.ottolenghi@sheba.health.gov.il (Y.W.-O.); noam.barda@sheba.health.gov.il (N.B.); maayan.gilboa@sheba.health.gov.il (M.G.); 2Infection Prevention & Control Unit, Sheba Medical Center, Tel Hashomer, Ramat Gan 52621, Israel; 3Sackler School of Medicine, Tel Aviv University, Ramat Aviv, Tel Aviv 6997801, Israel; elena.benshahar@sheba.health.gov.il (E.B.-S.); yitshak.kreiss@sheba.health.gov.il (Y.K.); yaniv.lustig@sheba.health.gov.il (Y.L.); 4Data Management Unit, Gertner Institute, Sheba Medical Center, Tel Hashomer, Ramat Gan 52621, Israel; carmitr@gertner.health.gov.il (C.R.); havim@gertner.health.gov.il (H.M.); 5Clinical Research, Joseph Sagol Neuroscience Center, Sheba Medical Center, Tel Hashomer, Ramat Gan 52621, Israel; 6Department of Software and Information Systems Engineering, Ben-Gurion University of the Negev, Be’er Sheva 8548800, Israel; 7Department of Epidemiology, Biostatistics and Community Health Sciences, Ben-Gurion University of the Negev, Be’er Sheva 8548800, Israel; 8ARC Innovation Center, Sheba Medical Center, Tel Hashomer, Ramat Gan 52621, Israel; 9General Management, Sheba Medical Center, Tel Hashomer, Ramat Gan 52621, Israel; 10Central Virology Laboratory, Public Health Services, Ministry of Health, Tel Hashomer, Ramat Gan 52621, Israel; viki.indenbaum@sheba.health.gov.il; 11Public Health Services, Ministry of Health, Jerusalem 9101002, Israel; sharon.alroy@moh.gov.il

**Keywords:** Long COVID, persistent long COVID symptoms, predictors, SARS-CoV-2 RBD-IgG, serum Neurofilament Light chain (NfL)

## Abstract

Background/Objectives: Millions of individuals worldwide continue to experience symptoms following SARS-CoV-2 infection. This study aimed to assess the prevalence and phenotype of multi-system symptoms attributed to Long COVID—including fatigue, pain, cognitive-emotional disturbances, headache, cardiopulmonary issues, and alterations in taste and smell—that have persisted for at least two years after acute infection, which we define as “persistent Long COVID”. Additionally, the study aimed to identify clinical features and blood biomarkers associated with persistent Long COVID symptoms. Methods: We sent a detailed long COVID symptoms questionnaire to an existing cohort of 1258 vaccinated adults (age 18–79 years) who had mild infection (e.g., non-hospitalized) SARS-CoV-2 Delta variant 2 years earlier. These individuals had comprehensive datasets, including blood samples, available for further analysis. We estimated prevalence of persistent long COVID two years post-infection using weighted adjustment (Horvitz–Thompson estimator) to overcome reporting bias. Multivariable logistic regression models were used to determine association of clinical features and blood biomarkers (pre-infection SARS-CoV-2 RBD-IgG, SARS-CoV-2 neutralizing antibodies, and pre-infection and post-infection neurofilament light) with prevalence of persistent long COVID. Results: N = 323 participants responded to the survey, of whom N = 74 (23%) reported at least one long COVID symptom that had persisted for two years after the acute infection. Weighted prevalence of persistent long COVID symptoms was 21.5% (95% CI = 16.7–26.3%). Female gender, smoking, and severity of acute COVID-19 infection were significantly associated with persistent Long COVID. The blood biomarkers assessed were not significantly associated with persistent Long COVID. Conclusions: Among vaccinated adults two years after mild infection with Delta variant SARS-CoV-2, persistent symptoms attributed to Long COVID are extremely common, certain subgroups are at higher risk, and further research into biological mechanisms and potential treatment targets is needed.

## 1. Introduction

Long COVID continues to impact the well-being of millions of COVID-19 recoverees worldwide who suffer from symptoms involving multiple organs and body systems [1,2]. The most common reported symptoms are fatigue, dyspnea, concentration problems, decreased exercise capacity, and sleep disorders [2,3,4]. These symptoms can be divided into symptom patterns according to the affected organ/system [5]. Most prior studies of long COVID symptoms have reported follow-up of one year or less after acute SARS-CoV-2 infection [1,2,3,4,6] and have reported prevalence rates of 7–30% in non-hospitalized patients. Here, we report long COVID symptoms in non-hospitalized patients two years after infection, which we define as “persistent long COVID”.

Given the very high reported prevalence of long COVID, there is an urgent need to identify those at highest risk to advance research on pathophysiology, treatment, and prevention. Several prior studies with heterogeneous sampling and time frames, patient populations, and SARS-CoV-2 variants have identified risk factors for long COVID, such as SARS-CoV-2 antibody response, immune and inflammatory response and comorbidities [7,8,9,10]. The pathophysiology of long COVID, however, remains poorly understood [11]. Moreover, most studies conducted, thus far, have not incorporated blood biomarkers into their assessment.

We hypothesized that antibodies directed against SARS-CoV-2 would protect against persistent long COVID by reducing the severity of acute infection. Prior studies assessing serum IgG levels among recoverees during the early months of the pandemic reported conflicting results [9,12,13,14]. Here, we tested the association between antibody levels in the pre- or early acute infection and persistent long COVID.

We further hypothesized that the serum neurofilament light chain (NfL) level in the pre-/early acute infection phase would be specifically associated with persistent long COVID cognitive-emotional symptoms. NfL is a biomarker of neuronal injury that has been reported to be associated with early post-COVID cognitive-emotional symptoms [15,16,17]. Cognitive-emotional symptoms are among the most commonly reported symptoms of long COVID and are hypothesized to result from neuroinflammation and abnormal neuroimmune responses [2,18,19,20,21]. Here, we tested the association between serum NfL levels in the pre-/early acute infection and persistent cognitive-emotional long COVID.

## 2. Methods

### 2.1. Study Design and Population

This prospective cohort study was nested within the Israeli COVID-19 Family Study (I-CoFS), a cohort study carried out during the Delta surge of the COVID-19 pandemic in Israel (25 July to 15 November 2021) [22]. Households of a new SARS-CoV-2 index case were approached for SARS-CoV-2 PCR and serology testing. Demographics, comorbidities, and information concerning the acute infection were collected for index cases and their households (Appendix A). During August–September 2023, two years after the acute infection, we contacted all positive cases (N = 1258) via a text message or a telephone call and invited them to answer a questionnaire concerning long COVID symptoms (Appendix A). Long COVID symptoms were divided into 8 patterns according to previous publications [5]: fatigue, pain, cognitive-emotional, pulmonary, cardiac, headache, taste and smell and other symptoms (see Appendix A for the detailed list of symptoms in each pattern). The study questionnaire included questions concerning 20 different symptoms. Participants were asked to report whether they suffered from these symptoms at any stage and for how long. Whenever a participant reported a symptom, he/she was requested to rank its impact on daily life using a Likert scale 0–4 (0—no impact at all; 4—maximal impact). In accordance with the World Health Organization (WHO) definition, a symptom was classified as long COVID only if it was new, persisted beyond 3 months following COVID-19, and lasted for at least 2 months [23]. Persistent long COVID is defined here as symptoms reported to last for over two years. For prevalence of long COVID, the different patterns and prevalence of persistent long COVID were calculated. The severity of COVID-19 acute infection symptoms was measured and categorized as negligible, mild, moderate and severe, as described in Regev-Yochay et al. [22].

The clinical predictors assessed included: age, sex, comorbidities, Body Mass Index (BMI), smoking habits, and last vaccine dose. The blood-biomarker predictors assessed included: pre-infection SARS-CoV-2-RBD-IgG antibody titer and neutralizing antibodies (a measure of pre-infection immune status), pre-infection serum NfL, and post-infection serum NfL at 3–4 weeks after infection.

### 2.2. Laboratory Tests

#### 2.2.1. Serology and Neutralization Assays

Blood samples were tested using the SARS-CoV-2 IgG II Quant (Abbott, IL, USA) test according to the manufacturer’s instructions and expressed as Binding Antibody Units (BAU) per the WHO standard. SARS-CoV-2 Pseudo-virus (psSARS-2) Neutralization Assay was performed as previously described [24].

#### 2.2.2. A Nested Case-Control Analysis of Serum NfL as a Predictor of Cognitive-Emotional-Long COVID

To determine whether serum NfL in the pre-infection (day 1 of infection) or early-infection period is associated with persistent cognitive-emotional-long COVID, we assessed serum NfL levels detected in the early acute COVID infection among individuals suffering from the cognitive-emotional long COVID pattern and compared them to a matched group without long COVID (see Section 2.3).

In addition, to determine whether an increase in serum NfL within the first month of the acute infection predicts persistent cognitive-emotional long COVID, we tested paired samples of five subjects experiencing persistent cognitive-emotional long COVID symptoms: on day 1 of infection (pre-infection) with their sample from approximately one month after infection (post-infection). We compared them to five subjects without any long COVID symptoms. The assay was performed as previously described [25].

### 2.3. Statistical Analysis

To estimate the true prevalence of long COVID, considering the high rate of non-responders (75%), we used the Horvitz–Thompson estimator, weighing adjustment methods using the propensity cell method [26]. First, a logistic propensity score (PS) model was applied to all the subjects, using the response indicator variable as the outcome and auxiliary variables, i.e., demographical characteristics, severity of COVID, and comorbidity (1+ vs. 0) as the predictors (Appendix A). From the predicted probabilities to respond we formed five groups (cells) according to their level, assuming the respondents within each cell are a random sample (i.e., assuming respondents and non-respondents have similar long COVID rates within each cell; in other words missing at random). Then, adjusting weights were calculated for subjects in each cell (i.e., number of subjects divided by number of respondents at each cell). In addition, considering potential unmeasured confounding that could not be estimated by the Horvitz–Thompson estimator, we considered an extreme scenario assuming that all non-responders were free of long COVID symptoms at the time of assessment, and calculated the proportion of persistent long COVID out of the entire target population.

To identify predictors of long COVID and persistent long COVID, we compared those with and without long COVID and those with and without persistent long COVID initially using the Mann–Whitney or Chi-square tests, as applicable, followed by a multivariable logistic model. All variables found significant at the level of 20% (*p* < 0.2) in the univariate analysis were considered as candidates for the multivariable logistic model. The variables sex, gender and IgG titers were forced into the multivariable model. To assess the humoral response, antibody levels were treated as categorical variables based on cutoffs that have been previously established [21]. A backward procedure was used to eliminate non-significant variables (*p* > 0.05) from the multivariable model. Odds ratios (OR) and 95% confidence intervals (95% CI) were reported. Analyses were implemented using the Statistical Analysis System (SAS).

### 2.4. Ethical Considerations

The Institutional Review Board of the Sheba Medical Center (SMC-8130-21) approved the protocol. All study participants signed a written informed consent.

Role of the funding source: The funding source was the Israeli Ministry of Health and internal SMC funding.

## 3. Results

Of the 1258 individuals (within the Israeli COVID-19 Family Study (I-CoFS)), infected with COVID-19 during the delta surge who were contacted, 323 (25.7%) agreed to participate in the study (i.e., respondents). Among respondents, 121/323 (37.5%) reported long COVID symptoms (Figure 1) and 74/323 (22.9%) fulfilled persistent long COVID criteria at two years following acute infection. Their characteristics are detailed in Table 1. Considering a non-response rate of 75% (935/1259), the analysis comparing various variables between respondents and non-respondents revealed that, apart from a statistically significant difference in age (with no biologic significance, i.e., 2.7 years) and BMI, no other differences were observed (Appendix A). The Horvitz–Thompson estimator weighted prevalence in the target population was 35.7% (95% CI = 30.0–41.4%) for long COVID and 21.5% (95% CI = 16.7–26.3%) for persistent long COVID. In the extreme scenario that all non-responders were free of long COVID symptoms, the prevalence of long COVID would be 9.6% (121/1258) and the prevalence of persistent long COVID would be 5.9% (74/1258).

Overall, 47% of individuals with persistent long COVID experienced more than three distinct symptom patterns. Among them, 72% reported fatigue-related symptoms, 43% reported cognitive-emotional symptoms, and 38% reported symptoms related to pain. (Figure 2, Table 2).

We further assessed the impact of symptom patterns on daily living. The most impactful symptom pattern was fatigue, followed by the cognitive-emotional symptoms pattern, and pain pattern (Table 2). Out of the 121 suffering from long COVID symptoms, 83 reported that at least one of the symptoms severely impacted their daily functioning.

In the multivariable logistic regression analysis (Table 3), females (OR = 2.2; 95% CI = 1.2–4.0), smokers (OR = 3.4; 95% CI = 1.9–6.1), and those with more severe symptoms during acute COVID-19 infection (OR = 3.4; 95% CI = 1.5–7.7) were more likely to report persistent long COVID. Pre-infection SARS-CoV-2-RBD IgG and neutralizing antibody levels or immediate post-infection levels were not associated with persistent long COVID. Since 92% of the cohort received two doses of vaccines against SARS-CoV-2, the number of vaccines was not included in the final model. Univariate analysis is reported in Appendix A.

Serum NfL levels were not significantly different between individuals with persistent cognitive-emotional long COVID and those without long COVID (Appendix A). Among the former, serum NfL levels during acute COVID-19, and 3–4 weeks later were also not significantly different (Appendix A).

## 4. Discussion

In this study following a cohort of individuals who contracted mild COVID-19 during the Delta variant era, we found that long COVID continues to have a significant impact. While 10–37% of the cohort reported long COVID of varied durations, at 2 years following the acute illness, 6–23% reported persistent long COVID. These results are in accordance with other studies [1,27,28,29] reporting a long COVID prevalence of 7–30% in non-hospitalized patients up to one year post-infection. Our study builds upon this prior work by reporting the prevalence of persistent long COVID at least two years post-infection.

Most previous studies report long COVID symptoms between 6- and 12-months post-infection, with fatigue being the most prevalent, followed by cognitive-emotional symptoms [1,2,3,4,12,13,14]. Our study identifies a subpopulation where long COVID persists for up to two years, with fatigue and cognitive-emotional symptoms remaining the most prevalent patterns. Moreover, our study demonstrates that even among individuals who did not require hospitalization for COVID-19, the severity of acute COVID-19 symptoms may be an independent predictor of persistent long COVID symptoms. Our finding that more severe acute disease is associated with higher risk for long COVID aligns with other recent high-quality studies [1,6] and provides further evidence debunking preliminary misperception that milder disease is associated with higher risk for long COVID [30,31]. The confirmation of this association in our study of only non-hospitalized patients is particularly notable as prior studies have included high proportions of hospitalized patients with much more severe disease.

Most studies investigating long COVID symptoms have focused on either the early stages of the pandemic or the Omicron variant era [29,32,33,34,35,36]. Studies that compared between the variants have reported a lower prevalence of long COVID following infection with the Omicron variant compared to the Delta variant [1,37,38,39]. Furthermore, acute COVID-19 during the Delta variant period was associated with a higher rate of lower respiratory tract involvement and more severe disease than the acute COVID-19 during the Omicron variant [6,37]. As a result, the higher prevalence of long COVID observed in our study and others may be linked to the specific characteristics of the Delta variant [37,38].

Previously reported predictors of long COVID 3–18 months post-infection include female gender, smoking, and high burden of symptoms during the acute phase [1,2,3,28,29]. Our study confirms these high-risk sub-groups and specifically reports that compared to men, women have more than twice the odds of persistent long COVID; compared to non-smokers, smokers have more than three times the odds of persistent long COVID; and compared to asymptomatic individuals with COVID-19, those with mild and moderate-severe symptoms of infection have more than twice the odds and more than 3 times the odds, respectively, of persistent long COVID.

Additionally, our study also tested blood biomarkers as potential predictors of persistent long COVID, including pre-infection SARS-CoV-2 antibodies and neutralizing antibodies. Previous studies assessing the association between antibody levels and long COVID, reported conflicting results [3,8,11,12,13]. However, these studies measured humoral response 1–18 months post infection. By contrast, we assessed peri-infection antibody levels, at an earlier stage of the initial disease, before or during acute infection. We hypothesized that higher peri-infection antibody levels could protect from long COVID, since neutralizing antibodies are known correlates of protection from SARS-CoV-2 infection and of COVID-19 severity [22,40,41,42,43]. Nonetheless, we did not find any association between either IgG or neutralizing antibody levels and long COVID or persistent long COVID. This negative finding may either reflect lack of association or merely point out that the interplay between the humoral response and long COVID is much more complex. Since we did not measure follow-up serological assays, it is possible that diminished humoral response following the infection or short-lived response did predict long COVID and/or persistent long COVID. Additionally, since almost the entire cohort received two vaccine doses, we could not examine an association between vaccination and long COVID.

Since the cognitive-emotional pattern is the most common pattern reported among individuals suffering from long COVID, we tested serum NfL, a recognized biomarker of neuronal injury during the acute phase, as a potential predictor. Previous research has indicated elevated levels of serum NfL in individuals at different time points (1 week to a few months) following SARS-CoV-2 infection [16,17,19,44]. We did not identify a significant difference in pre-infection serum NfL concentrations between individuals with or without persistent cognitive-emotional long COVID. Similarly, pre-infection and post-infection (3–4 weeks after acute infection) serum NfL concentrations did not significantly differ between the groups.

Our analysis was underpowered and does not rule out neural injury as an explanatory mechanism or autoimmune processes (e.g., bioactive GPCR auto-antibodies; microthrombotic processes etc.) with reduction of oxygen saturation of the tissue. Moreover, cognitive-emotional long COVID could alternatively stem from autonomic dysfunction or emotional distress [39,45]. These findings are preliminary, and further research is required to establish any conclusive evidence.

Our study has several limitations. First, there is potential for recall bias and response bias. While we acknowledge that the very high non-response rate substantially reduces the sample size, increases the risk of bias, and reduces the power of the reported results, we have maximally mitigated these limitations by adjusting for potential differences in the characteristics of responders and non-responders and by considering the extreme scenario in which all non-responders were free from long COVID. Second, our estimate of the prevalence of long COVID may not generalize to non-Delta variants of SARS-CoV-2. Third, we measured the blood biomarkers during the pre-infection and only once post-infection (3–4 weeks after acute infection) without performing repeated follow-up measurement. Thus, differences in the humoral response and later serum NfL fluctuations might have been missed.

In conclusion, our study reveals that long COVID symptoms following the Delta variant surge occurred at a high rate and have persisted for more than 2 years at a non-negligible proportion. While the risk factors for persistent long COVID included female gender, smoking, and the severity of acute phase symptoms, we did not find an association between humoral immunity or serum NfL early during the course of illness with persistent long COVID. These findings underscore the complexity of characterizing long COVID and highlight the need for further research to elucidate mechanisms and to inform the targeted prevention and treatment of long COVID.

## Figures and Tables

**Figure 1 viruses-16-01955-f001:**
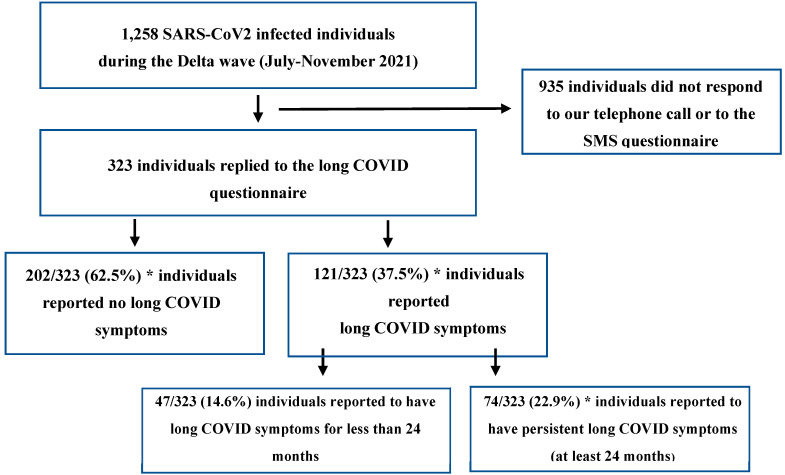
Study population and long COVID prevalence. * In a case where all 935 non-responders were free of long COVID symptoms, these rates would be 1137/1258 (90.4%) without long COVID and 121/1258 (9.6%) with long COVID and 74/1258 (5.9%) with persistent long COVID.

**Figure 2 viruses-16-01955-f002:**
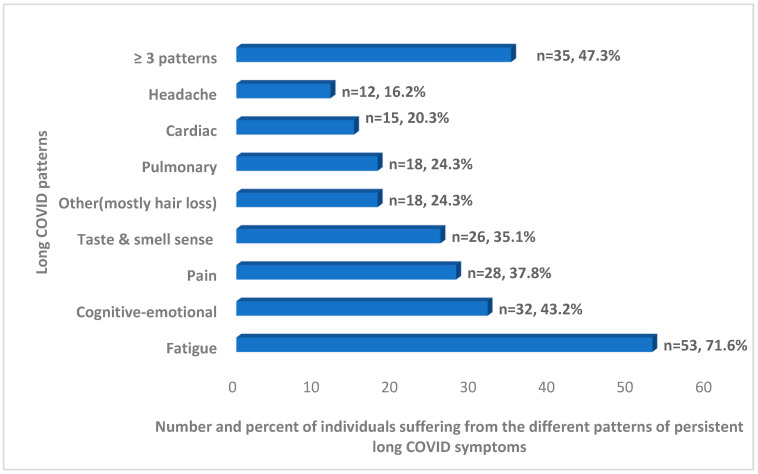
A proportion of the different patterns of long COVID symptoms from the subjects suffering from persistent long COVID (N/74, %).

**Table 1 viruses-16-01955-t001:** Population Characteristics.

Characteristic N(%) Unless Otherwise Specified	All	No Long COVID	Long COVID	Persistent Long COVID	*p*-Value
					(No Long COVID vs. Persistent Long COVID)
**N**	**323**	**202/323** **62.5%**	**121/323** **37.5%**	**74/323** **22.9%**	
Age, years, Median (IQR)	46.0 (37.0–55.0)	45.0 (36.0–54.0)	47.0 (38.0–55.0)	47.5 (39–57)	0.2135
Female	189 (58.5%)	109 (54.0%)	80 (66.1%)	51 (68.9%)	0.0257
1+ comorbidities	86 (26.6%)	48 (23.8%)	38 (31.4%)	24 (32.4%)	0.1462
BMI					0.6155
<18	141 (44.3%)	93 (47.0%)	48 (40.0%)	28 (38.4%)	
18–24.9	111 (34.9%)	69 (34.9%)	42 (35.0%)	29 (38.4%)	
25–29.9	48 (15.1%)	26 (13.1%)	22 (18.3%)	30 (38.4%)	
>29.9	18 (5.7%)	10 (5.1%)	8 (6.7%)	31 (38.4%)	
Unknown	5 missing	4 missing	1 missing	32 (38.4%)	
Smoking	122 (37.8%)	64 (31.7%)	58 (47.9%)	33 (38.4%)	0.0007
Number of vaccines					0.6662
0	13 (4.0%)	9 (4.5%)	4 (3.3%)	2 (2.7%)	
1	1 (0.3%)	1 (0.5%)	0 (0.0%)	0 (0.0%)	
2	309 (95.7%)	192 (95.1%)	117 (96.7%)	7 (97.3%)	
Severity of COVID-19 infection					0.0231
Asymptomatic/negligible	83 (25.8%)	62 (30.9%)	21 (17.4%)	12 (16.2%)	
Mild	149 (46.3%)	89 (44.3%)	60 (49.6%)	34 (46.0%)	
Moderate & severe	90 (28.0%)	50 (24.9%)	40 (33.1%)	28 (37.8%)	

**Table 2 viruses-16-01955-t002:** Frequency and Severity of reported long COVID symptoms by patterns among respondents reporting long COVID (N = 121) and persistent long COVID (N = 74).

Total (N = 323)	Fatigue	Pain	Cognitive-Emotional	Pulmonary	Headache	Cardiac	Taste and Smell	Other
	N (%)	N (%)	N (%)	N (%)	N (%)	N (%)	N (%)	N (%)
**Symptom pattern**	88/323 (27.2%)	50/323 (15.5%)	54/323 (16.7%)	32/323 (9.9%)	22/323 (6.8%)	30/323 (9.3%)	47/323 (14.6%)	32/323 (9.9%)
**Impact of Symptom on daily function ***								
No impact	1 (1.1%)	5 (10.0%)	3 (5.6%)	2 (6.3%)	2 (9.1%)	4 (13.3%)	6 (12.8%)	2 (6.3%)
Mild to moderate impact	54 (61.4%)	22 (44.0%)	21 (38.9%)	11 (34.4%)	13 (59.1%)	17 (56.7%)	24 (51.1%)	12 (37.5%)
Severe impact	31 (35.2%)	22 (44.0%)	28 (51.9%)	18 (56.3%)	5 (22.7%)	5 (16.7%)	16 (34.0%)	16 (50.0%)
missing	2 (2.3%)	1 (2.0%)	2 (3.7%)	1 (3.1%)	2 (9.1%)	4 (13.3%)	1 (2.1%)	2 (6.3%)
**Symptom pattern duration**								
<6 months	29 (33.0%)	15 (30.0%)	21 (38.9%)	20 (62.5%)	9 (40.9%)	15 (50.0%)	21 (44.7%)	12 (37.5%)
6–24 months	7 (8.0%)	2 (4.0%)	21 (38.9%)	1 (3.1%)	0	2 (6.7%)	7 (14.9%)	5 (5.6%)
persistent long COVID (>24 months)	52 (59.1%)	33 (66.0%)	12 (22.2%)	11 (34.4%)	13 (59.1%)	12 (40.0%)	19 (40.4%)	15 (46.9%)

* Impact of Symptom on daily function: No impact—the symptom did not interfere with daily functioning.

**Table 3 viruses-16-01955-t003:** Multivariable logistic regression identifying independent predictors of persistent long COVID.

	Value	OR	Lower	Upper	*p*-Value
Age		1.0	0.988	1.029	0.415
Gender (REF: Men)	Women	2.2	1.208	4.003	0.01
Smoking (REF: non-smoker)	Smoker	3.4	1.847	6.075	<0.0001
Severity of acute COVID-19 infection (REF: asymptomatic/negligible)					
	Mild	2.0	0.92	4.184	0.081
	Moderate/severe	3.4	1.523	7.663	0.003
Pre-infection RBD- IgG (REF: <300)					
	300–500	1.2	0.484	3.077	0.674
	>500	1.6	0.779	3.424	0.194

## Data Availability

All data will be available upon request.

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
