# Peer review of "Persistence of Long COVID Symptoms Two Years After SARS-CoV-2 Infection: A Prospective Longitudinal Cohort Study"

_viruses, 2024, doi:10.3390/v16121955_

Round 1

Reviewer 1 Report

Comments and Suggestions for Authors

Despite numerous publications, the prevalence of LongCOVID (LC) remains unclear. In this regard, this study, which includes a considerable sample of patients, allows for an estimate of the prevalence. However, the high non-response rate (75%) significantly reduces the sample size, increases bias, and consequently lowers the power of the reported results. 

Regarding the clinical analysis a noteworthy finding is the association found by the authors between the severity of acute disease and the presence of LC. This contrasts with  clinical observations and findings from other studies, which report that LC is more common in patients with mild acute disease. I missed an explanation to this finding in the conclusions. 

In relation to laboratory studies aimed to identifying biomarkers related to LC, only five patients were included to evaluate the relationship between neurofilament light chain (NfL) levels and cognitive-emotional impairment. The number of patients in this subanalysis is very small, making it impossible to draw any meaningful conclusions. While the authors acknowledge this limitation, the extremely small sample size and the absence of serial measurements make these results of limited value. 

Author Response

Reviewer 1

We thank you for your prompt and constructive review. Below are the answers relating to all the comments:

Quality of English Language

(x) The quality of English does not limit my understanding of the research.
( ) The English could be improved to more clearly express the research.

Answer: The manuscript has now been thoroughly edited by a professional Native American English speaker.

  1. Despite numerous publications, the prevalence of LongCOVID (LC) remains unclear. In this regard, this study, which includes a considerable sample of patients, allows for an estimate of the prevalence. However, the high non-response rate (75%) significantly reduces the sample size, increases bias, and consequently lowers the power of the reported results. 

Answer: We agree with the reviewer. We have further expanded upon this limitation in the Discussion of limitations as follows (lines 263-268), “Our study has several limitations. First, there is potential for recall bias and response bias. While we acknowledge that the very high non-response rate substantially reduces the sample size, increases the risk of bias, and reduces the power of the reported results, we have maximally mitigated these limitations by adjusting for potential differences in the characteristics of responders and non-responders and by considering the extreme scenario in which all non-responders were free from long COVID"

  1. Regarding the clinical analysis a noteworthy finding is the association found by the authors between the severity of acute disease and the presence of LC. This contrasts with  clinical observations and findings from other studies, which report that LC is more common in patients with mild acute disease. I missed an explanation to this finding in the conclusions. 

Answer: Thank you for raising this important point which is a common source of confusion. We now clarify in our revised Discussion, that “Our finding that more severe acute disease is associated with higher risk for long COVID aligns with other recent high-quality studies (1,6) and provides further evidence debunking preliminary misperception that milder disease is associated with higher risk for long COVID (30, 31). The confirmation of this association in our study of only non-hospitalized patients is particularly notable as prior studies have included high proportions of hospitalized patients with much more severe disease.

Here are quotes from these studies:

"The incidence is estimated at 10-30% of non-hospitalized cases, 50-70%of hospitalized cases, and 10-12% of vaccinated cases."

  1. 1. Davis HE, McCorkell L, Vogel JM, Topol EJ. Long COVID: major findings, mechanisms and recommendations. Nat Rev Microbiol. 2023 Mar;21(3):133-146. doi: 10.1038/s41579-022-00846-2. Epub 2023 Jan 13. Erratum in: Nat Rev Microbiol. 2023 Jun;21(6):408. doi: 10.1038/s41579-023-00896-0. PMID: 36639608; PMCID: PMC9839201.

"The risk of PASC appears to increase with greater severity of infection…."

  1. 6. Xie Y, Choi T, Al-Aly Z. Postacute Sequelae of SARS-CoV-2 Infection in the Pre-Delta, Delta, and Omicron Eras. N Engl J Med. 2024 Aug 8;391(6):515-525. doi: 10.1056/NEJMoa2403211. Epub 2024 Jul 17. PMID: 39018527.

  1. In relation to laboratory studies aimed to identifying biomarkers related to LC, only five patients were included to evaluate the relationship between neurofilament light chain (NfL) levels and cognitive-emotional impairment. The number of patients in this subanalysis is very small, making it impossible to draw any meaningful conclusions. While the authors acknowledge this limitation, the extremely small sample size and the absence of serial measurements make these results of limited value.

Answer: In this cohort, only a small subset was available for studying NfL as a biomarker associated with long COVID. At no point did we intend to draw definitive conclusions from these results. To clarify this, we have added a statement to the discussion emphasizing that these findings are preliminary, and that further research is required to establish any conclusive evidence.

We added a sentence to the discussion:” Our analysis was under-powered and does not rule out neural injury as an explanatory mechanism or autoimmune processes (e.g. bioactive GPCR auto-antibodies; microthrombotic processes etc.) with reduction of oxygen saturation of the tissue. Although, cognitive-emotional long COVID could alternatively stem from autonomic dysfunction or emotional distress (39, 45). These findings are preliminary, and further research is required to establish any conclusive evidence.

Reviewer 2 Report

Comments and Suggestions for Authors

Congratulations to authors for the work that you have conceived and carried out; it was a pleasure to read. It is highly interesting and suitable not only for readers and colleagues interested in virology but also for clinicians from various specialties and epidemiologists. The study is rigorous, the questionnaire is precise, and the discussion is thorough.

The only suggestion (minor revisions) concerns the title, where the phrase “longitudinal study” appears too generic. I would recommend adding a phrase that hints at the type of study conducted and/or the results obtained.

Comments on the Quality of English Language

The quality of English is fine

Author Response

Reviewer 2

We thank you for your prompt and constructive and warm review. Below are the answers relating to all the comments:

  1. The English could be improved to more clearly express the research.

 Answer: The manuscript has now been thoroughly edited by a professional Native American English speaker.

  1. Congratulations to authors for the work that you have conceived and carried out; it was a pleasure to read. It is highly interesting and suitable not only for readers and colleagues interested in virology but also for clinicians from various specialties and epidemiologists. The study is rigorous, the questionnaire is precise, and the discussion is thorough.

The only suggestion (minor revisions) concerns the title, where the phrase “longitudinal study” appears too generic. I would recommend adding a phrase that hints at the type of study conducted and/or the results obtained.

Answer: We appreciate this comment and have now changed the title accordingly to give more information on the type of study. The Title is now:

Persistence of Long COVID symptoms two years after SARS-CoV-2 infection: a prospective longitudinal cohort study

Reviewer 3 Report

Comments and Suggestions for Authors

The work is clearly presented in its objective of identifying biomarkers and risk factors for long COVID. The statistics used are adequate and critically applied. The discussion is self-critical and concise.

There are only a few minor comments to be made:

On page 4 in Fig. 1, the time could be added to the bottom box for clarity: 47/323 (14.6%) individuals reported to have long COVID symptoms for less than 24 months.

In the discussion on the question of brain fog or neurological symptoms in Long Covid, it should at least be mentioned that not only neurodegeneration, as the NfL as a marker is one possibility and in the work it does not turn out to be a risk factor, but also autoimmune processes (e.g. bioactive GPCR auto-antibodies; microthrombotic processes etc.) with reduction of oxygen saturation of the tissue. 

Author Response

Reviewer 3

We thank you for your prompt and constructive and warm review. Below are the answers relating to all the comments:

(x) The quality of English does not limit my understanding of the research.

 Answer: The manuscript has now been thoroughly edited by a professional Native American English speaker.

  1. The work is clearly presented in its objective of identifying biomarkers and risk factors for long COVID. The statistics used are adequate and critically applied. The discussion is self-critical and concise.There are only a few minor comments to be made:On page 4 in Fig. 1, the time could be added to the bottom box for clarity: 47/323 (14.6%) individuals reported to have long COVID symptoms for less than 24 months.

Answer: This was corrected: the word “months” was added

  1. In the discussion on the question of brain fog or neurological symptoms in Long Covid, it should at least be mentioned that not only neurodegeneration, as the NfL as a marker is one possibility and in the work it does not turn out to be a risk factor, but also autoimmune processes (e.g. bioactive GPCR auto-antibodies; microthrombotic processes etc.) with reduction of oxygen saturation of the tissue. 

Answer: Thank you for your comment. We have added a sentence in the discussion (lines 255-259):" Our analysis was underpowered and does not rule out neural injury as an explanatory mechanism or autoimmune processes (e.g. bioactive GPCR auto-antibodies; microthrombotic processes etc.) with reduction of oxygen saturation of the tissue. Although, cognitive-emotional long COVID could alternatively stem from autonomic dysfunction or emotional distress (37, 43).

Round 2

Reviewer 1 Report

Comments and Suggestions for Authors

After the review and changes made by the authors, the limitations identified in this study have been clearly improved. I believe the overall quality of the study has been enhanced, and its limitations are now more clearly outlined